# Low-Grade Endometrial Cancer with Abnormal p53 Expression as a Separate Clinical Entity: Insights from RNA Sequencing and Immunohistochemistry

**DOI:** 10.3390/diagnostics15060671

**Published:** 2025-03-10

**Authors:** Kazuhisa Hachisuga, Minoru Kawakami, Hiroshi Tomonobe, Shoji Maenohara, Keisuke Kodama, Hiroshi Yagi, Masafumi Yasunaga, Ichiro Onoyama, Kazuo Asanoma, Hideaki Yahata, Yoshinao Oda, Kiyoko Kato

**Affiliations:** 1Department of Gynecology and Obstetrics, Graduate School of Medical Sciences, Kyushu University, Fukuoka 812-8582, Japan; kawakami.minoru.232@m.kyushu-u.ac.jp (M.K.); tomonobe.hiroshi.204@m.kyushu-u.ac.jp (H.T.); maenohara.shoji.001@m.kyushu-u.ac.jp (S.M.); kodama.keisuke.839@m.kyushu-u.ac.jp (K.K.); yagi.hiroshi.975@m.kyushu-u.ac.jp (H.Y.); yasunaga.masafumi.983@m.kyushu-u.ac.jp (M.Y.); onoyama.ichiro.231@m.kyushu-u.ac.jp (I.O.); asanoma.kazuo.992@m.kyushu-u.ac.jp (K.A.); yahata.hideaki.134@m.kyushu-u.ac.jp (H.Y.); kato.kiyoko.172@m.kyushu-u.ac.jp (K.K.); 2Department of Anatomic Pathology, Graduate School of Medical Sciences, Kyushu University, Fukuoka 812-8582, Japan; oda.yoshinao.389@m.kyushu-u.ac.jp

**Keywords:** low-grade endometrial cancer, p53, L1CAM, p21, molecular classification

## Abstract

**Background:** A molecular classification of endometrial cancer was developed based on an analysis of The Cancer Genome Atlas. In this classification, the group characterized by abnormal p53 immunohistochemical expression showed the poorest prognosis. However, there may be no need to apply a molecular classification in low-grade endometrial cancer. In this study, we investigated the clinical significance of abnormal p53 immunohistochemical expression in low-grade endometrial cancer. **Methods:** We obtained nine frozen samples of endometrial cancer [low-grade endometrial cancer with wild-type p53 expression (EC^lo^p53^wt^ group): n = 3, low-grade endometrial cancer with abnormal p53 expression (EC^lo^p53^ab^ group): n = 3, and high-grade endometrial cancer (EC^hi^ group): n = 3]. RNA sequencing was performed for each sample. All the samples passed RNA quality control. In addition, an immunohistochemical analysis was performed for 44 formalin-fixed paraffin-embedded samples. **Results:** Differentially expressed genes were identified in the RNA sequencing results (1811 genes between the EC^lo^p53^ab^ group and the EC^hi^ group, and 1088 genes between the EC^lo^p53^ab^ group and the EC^lo^p53^wt^ group). In a principal component analysis, the EC^lo^p53^ab^ group was more similar to the EC^lo^p53^wt^ group than to the EC^hi^ group. In the immunohistochemical analysis, L1CAM expression was significantly less frequently observed in the EC^lo^p53^ab^ group than in the EC^hi^ group. Moreover, p21 expression tended to be more frequently observed in the EC^lo^p53^ab^ group than in the EC^hi^ group. **Conclusions:** In this study, the RNA sequencing and immunohistochemical results revealed that the EC^lo^p53^ab^ group is a separate entity from the EC^hi^ group. While the abnormal p53 group is considered the most prognostically unfavorable in molecular classification, these findings suggest that routine molecular profiling is not necessary for patients with low-grade endometrial cancer. However, there is insufficient evidence to modify adjuvant treatment in low-grade endometrial cancer patients. Further investigation is needed on the clinical application of molecular classification to low-grade endometrial cancer.

## 1. Introduction

Endometrial cancer is divided into type 1 and type 2, according to the Bokhman classification, published in 1983 [1]. To update this, in 2013, The Cancer Genome Atlas (TCGA) Research Network conducted a comprehensive genomic and transcriptomic analysis of endometrial cancers. Based on this analysis, endometrial cancers were classified into four genomic subtypes: (1) *POLE* ultramutated tumors, which are characterized by hotspot mutations in the exonuclease domain of *POLE* (a subunit of DNA polymerase ε involved in DNA replication) and extremely high mutation rates; (2) microsatellite-unstable tumors, marked by microsatellite instability due to the dysfunction of the DNA mismatch repair (MMR) proteins; (3) copy-number-low tumors, predominantly comprising low-grade endometrioid cancers with low mutation rates and microsatellite stability, and typically featuring frequent *CTNNB1* mutations; and (4) copy-number-high tumors, characterized by extensive copy number aberrations, recurrent *TP53* mutations, and low mutation rates [2]. Among these subtypes, copy-number-high tumors are associated with the poorest prognosis.

Building upon these findings, Kommoss et al. proposed a molecular classification of endometrial cancer, referred to as the Proactive Molecular Risk Classifier for Endometrial Cancer [3]. This classification has since been integrated into the ESGO/ESTRO/ESP guidelines of 2021 and the FIGO 2023 staging system [4,5]. The usefulness of molecular classification for high-grade endometrial cancer has been reported, but there is insufficient evidence of the value of applying it for low-grade endometrial cancer [4]. In addition, Vrede et al. reported that low-grade endometrial cancer has an excellent prognosis, regardless of its molecular subgroup [6].

Abnormal p53 expression is a rare finding in low-grade endometrial cancer, being reported in 2–15% of cases [3,4,5,6,7,8,9,10,11,12]. In our previous study, we reported that abnormal p53 immunohistochemical expression was not an independent prognostic factor when confined to endometrioid carcinoma, G1 [13]. Meanwhile, Puppo et al. demonstrated conflicting findings that low-grade endometrial cancer patients with abnormal p53 expression seem to be at a greater risk of recurrence [12]. When molecular classification is applied to low-grade endometrial cancer, there is a possibility that adjuvant therapy may be modified for the abnormal p53 group. However, there is still insufficient evidence to support this. The question of whether to evaluate p53 expression in all cases of low-grade endometrial cancer also warrants further investigation. Against this background, we conducted the present study to reveal the clinical significance of abnormal p53 immunohistochemical expression in low-grade endometrial cancer.

## 2. Materials and Methods

### 2.1. Case Selection and Clinicopathological Characteristics

All the samples used in this study were collected at the Department of Gynecology and Obstetrics, Kyushu University, from 1992 to 2022. We obtained nine frozen samples of endometrial cancer [low-grade endometrial cancer with wild-type p53 expression (EC^lo^p53^wt^ group): n = 3, low-grade endometrial cancer with abnormal p53 expression (EC^lo^p53^ab^ group): n = 3, and high-grade endometrial cancer (EC^hi^ group): n = 3]. In this study, the two EC^lo^ groups were histologically confirmed to involve endometrioid carcinoma grades 1–2 (G1–2) and the EC^hi^ group was characterized by severe nuclear atypia, the papillary structure, abnormal p53 expression, and decreased ER expression, with histologically and immunohistochemically confirmed conventional uterine corpus serous carcinoma. In molecular classification, uterine corpus serous carcinoma was reported to be mostly classified in the abnormal p53 group. RNA sequencing was performed for each sample. 

Next, we reviewed the medical records of 44 patients with histologically and immunohistochemically confirmed endometrial cancer, categorized into three groups (EC^lo^p53^wt^ group: n = 16, EC^lo^p53^ab^ group: n = 13, and EC^hi^ group: n = 15). Clinical and pathological data were extracted from the records and hematoxylin and eosin (HE)-stained slides, including information on the International Federation of Gynecology and Obstetrics (FIGO) 2023 stage, age, metastasis, depth of myometrial invasion, lymphovascular invasion, peritoneal cytology, uterine cervical invasion, therapeutic methods, recurrence, progression-free survival (PFS), and overall survival (OS).

For the age category, patients aged 60 years or older were classified as elderly. Lymph node metastasis was further subclassified based on the amount of tumor tissue, as follows: isolated tumor cells (ITCs) (≥0.2 mm and ≤200 cells), micro-metastasis (0.2–2 mm and/or >200 cells), and macro-metastasis (>2 mm) [14]. Lymphovascular invasion was classified as none, focal, or substantial. Substantial lymphovascular invasion was defined as the involvement of ≥5 vessels, while focal lymphovascular invasion was defined as the involvement of <5 vessels [15].

All the methods were carried out in accordance with relevant guidelines and regulations. Informed consent was obtained from all the participants. The institutional review board of Kyushu University approved this study (project number: 23085-00).

### 2.2. RNA Sequencing and Data Processing

#### 2.2.1. Total RNA Isolation

The total RNA was isolated from endometrial cancer tissue using the TRIzol reagent (Invitrogen, Waltham, MA, USA) and purified using an SV Total RNA Isolation System (Promega, Madison, WI, USA), in accordance with the manufacturer’s instructions. The RNA samples were quantified using an ND-1000 spectrophotometer (NanoDrop Technologies, Wilmington, DE, USA) and the quality was confirmed with a Tapestation (Agilent, Santa Clara, CA, USA). All the RNA used for sequencing was confirmed to have an RIN value of 7.0 or higher.

#### 2.2.2. RNA Sequencing 

Sequencing libraries were constructed from 200 ng of total RNA using an MGIEasy rRNA Depletion Kit and an MGIEasy RNA Directional Library Prep Set (MGI Tech Co., Ltd., Shenzhen, China), following the manufacturer’s protocol. The libraries were then sequenced on the DNBSEQ-G400 FAST Sequencer (MGI Tech Co., Ltd.) by employing a paired-end 150 nt strategy.

#### 2.2.3. Sequencing Data Analysis

The raw sequencing reads were processed to remove low-quality bases and adapters using Trimmomatic (v.0.38) [16]. Gene-level raw counts for each sample were estimated with RSEM version 1.3.0 and Bowtie 2 [17,18]. Differentially expressed genes (DEGs) were identified using the edgeR [19] program. For DEG detection, normalized counts per million (CPM), log fold changes (logFC), and *p*-values were calculated from the raw counts. DEGs were defined according to the following criteria: *p*-value < 0.05 and logFC ≥ 1 (for upregulated genes) or logFC ≤ −1 (for downregulated genes).

#### 2.2.4. Immunohistochemical Staining

Antibodies

The following primary antibodies were used: p53 (mouse monoclonal, clone DO-7, dilution of 1:100; Novocastra, Newcastle, UK), ER (mouse monoclonal, clone 6F11, dilution of 1:50; Novocastra), L1CAM (mouse monoclonal, clone 14.10, dilution of 1:50; BioLegend, San Diego, CA, USA), p21 (mouse monoclonal, clone 4D10, dilution of 1:20; Novocastra), PMS2 (mouse monoclonal, clone A16-4, dilution of 1:200; BD Biosciences, Heidelberg, Germany), and MSH6 (rabbit monoclonal, clone EP49, dilution of 1:200; Dako, Glostrup, Denmark).

Procedure

Four-micrometer-thick tissue sections on silane-coated slides were stained using the universal immunoperoxidase polymer method (Envision kit; Dako), following the manufacturer’s guidelines. After deparaffinization, rehydration, and the blocking of endogenous peroxidase activity, the sections were incubated with primary antibodies at room temperature for 90 min. Following incubation with the secondary antibody at room temperature, the sections were treated with 3,3¢-diaminobenzidine, counterstained with hematoxylin, and mounted. Antigen retrieval was achieved through microwave heating and pressure cooking. Positive and negative controls were also included in the procedure.

Immunohistochemical Scoring

Abnormal p53 expression was defined by either strong nuclear staining in at least 80% of the tumor cell nuclei or the complete absence of nuclear staining when the “wild-type” pattern was present, characterized by a range from a few positive cells to nearly all the cells being stained, albeit with a varying intensity [11,12]. The cytoplasmic pattern of p53 expression was characterized by predominant cytoplasmic staining without strong nuclear staining in more than 80% of tumor cell nuclei [12,20]. The assessment of immunohistochemical p53 expression was carried out by a pathologist and a physician specializing in gynecological oncology, both of whom were blinded to the clinicopathological details of the patients [11]. Regarding the L1CAM expression, such expression was scored according to the percentage of positivity in tumor cells (score 0 = 0%, score 1 = 1%–10%, score 2 =>10%–50%, and score 3 =>50%). If 10% or more of the tumor cells showed L1CAM staining, the cancer was defined as positive [21,22]. Immunohistochemical staining for p21 was scored as absent, focal, or diffuse. We defined focal and diffuse cases as those with positivity for p21 immunohistochemical expression [23]. The expression of PMS2 or MSH6 was considered “lost” when there was a complete absence of nuclear staining in tumor cells, while the normal surrounding cells consistently exhibited preserved nuclear staining.

### 2.3. Statistical Analysis

All the statistical analyses were conducted using the statistical software JMP® 16 (SAS Institute, Cary, NC, USA). Fisher’s exact test and Pearson’s chi-squared test were used for the data analysis. For the survival analysis, Kaplan–Meier curves were generated, and the log-rank test was applied. Univariate survival analyses were carried out using the Cox proportional hazards model. A *p*-value of <0.05 was considered statistically significant in all tests.

## 3. Results

### 3.1. Clinicopathological Characteristics

After thoroughly reviewing the clinical data of the 44 endometrial cancer cases, we reclassified them according to the FIGO 2023 staging system. The clinicopathological characteristics of these cases are summarized in Table 1.

In the EC^lo^p53^ab^ group, elderly individuals (≥60 years) accounted for 69.2% of the total cases. Deep myometrial invasion (≥1/2) was present in 30.8% of these patients. Lymphovascular invasion was detected in 23.1% of the patients (focal lymphovascular invasion = 2 and substantial lymphovascular invasion = 1), while lymph node metastasis was found in one patient (macrometastasis). Overall, 76.9% of the patients had early cancers and 23.1% had advanced cancers. As for treatment, all of the patients underwent a total hysterectomy and lymphadenectomy in this study. Three recurrent cases were observed here.

### 3.2. RNA Sequencing Analysis

MA plots of the RNA sequencing results comparing logFC and logCPM were created to identify specific genes that fit the threshold criteria (*p*-value < 0.05 and logFC ≥ 1 or ≤−1) (Figure 1). A volcano plot of the differential gene expression patterns of the EC^lo^p53^ab^ group versus the EC^hi^ group was also created (Figure 2). DEGs were identified in the RNA sequencing results (1811 genes between the EC^lo^p53^ab^ group and EC^hi^ group, 1088 genes between the EC^lo^p53^ab^ group and EC^lo^p53^wt^ group, and 2370 genes between the EC^lo^p53^wt^ group and EC^hi^ group). In the principal component analysis (PCA), the EC^lo^p53^ab^ group was more similar to the EC^lo^p53^wt^ group than to the EC^hi^ group, as shown in Figure 3.

### 3.3. Immunohistochemical Analysis 

In the RNA sequencing analysis comparing the EC^lo^p53^ab^ group to the EC^hi^ group, *L1CAM* was identified as a downregulated gene in the former group, while *CDKN1A* encoding p21 was upregulated in it. In addition, in the RNA sequencing analysis comparing the EC^lo^p53^ab^ group to the EC^lo^p53^wt^ group, *CDKN1A* was identified as a downregulated gene in the former group. Both L1CAM and p21 expression are associated with p53 expression. Therefore, we evaluated the L1CAM and p21 immunohistochemical expression in 44 samples (EC^lo^p53^wt^ group: n = 16, EC^lo^p53^ab^ group: n = 13, and EC^hi^ group: n = 15).

Representative HE-stained and immunohistochemical images are shown in Figure 4. The L1CAM immunohistochemical results are shown in Table 2.

In the EC^lo^p53^ab^ group, two cases (15.4%) were positive for L1CAM expression. In contrast, 14 cases (93.3%) were positive in the EC^hi^ group, while all the cases in the EC^lo^p53^wt^ group were negative for L1CAM expression. These results indicate a significant difference between the EC^lo^p53^ab^ group and the EC^hi^ group (*p* < 0.0001). Meanwhile, there was no significant difference in the L1CAM expression rate between the EC^lo^p53^ab^ group and EC^lo^p53^wt^ group (*p* = 0.1039).

Representative p21 immunohistochemical images are shown in Figure 5 and the p21 immunohistochemical results are shown in Table 3. 

In the EC^lo^p53^ab^ group, six cases (46.2%) were positive for p21 expression. In contrast, 13 (81.3%) cases were positive in the EC^lo^p53^wt^ group, while 4 cases (26.7%) were in the EC^hi^ group. These results indicate a significant difference between the EC^lo^p53^ab^ group and the EC^lo^p53^wt^ group (*p* = 0.0480). Meanwhile, there was no significant difference in the p21 expression rate between the EC^lo^p53^ab^ group and EC^hi^ group (*p* = 0.2831). In addition, there was a significant association between L1CAM and p21 expression in this study (*p* = 0.0348).

### 3.4. Survival Analysis

In this study, there were three cases of recurrence in the EC^lo^p53^ab^ group (n = 13). Meanwhile, seven such cases were observed in the EC^hi^ group (n = 12). The EC^lo^p53^ab^ group had a significantly longer progression-free and overall survival than the EC^hi^ group in the Kaplan–Meier analysis (*p* = 0.0023 and 0.0249, respectively), as shown in Figure 6. In addition, univariate analyses demonstrated that the hazard ratios for the PFS and OS in the EC^hi^ group versus the EC^lo^p53^ab^ group were 6.30 (95% confidence interval, 1.62–24.47) and 5.66 (95% confidence interval, 1.05–30.47), respectively.

## 4. Discussion

Endometrial cancers are divided into types 1 and 2, according to the classification proposed by Bokhman [1]. The representative histology of type 1 is endometrioid carcinoma and type 1 tumors have a more favorable prognosis than type 2 ones. More than 85% of patients with endometrial cancer present with low-grade histology (endometrioid carcinoma G1–2) [6].

Recently, a molecular classification for endometrial cancer has been developed and efforts have been made to apply it in clinical practice. However, the value of molecular classification for low-grade endometrial cancer is still unclear. Vrede et al. reported that low-grade endometrial cancer has an excellent prognosis independently of the molecular subgroup [6]. In addition, we previously reported that p53 immunohistochemical expression is not an independent prognostic factor, and is confined to endometrioid carcinoma, G1 [13].

Against this background, we conducted the present study to clarify the clinical significance of abnormal p53 expression in low-grade endometrial cancer. In the RNA sequencing analysis and PCA in this study, the EC^lo^p53^ab^ group was more similar to the EC^lo^p53^wt^ group than to the EC^hi^ group. In the RNA sequencing analysis, *L1CAM* was identified as being downregulated more frequently in the EC^lo^p53^ab^ group than in the EC^hi^ group.

The upregulation of *L1CAM* was found to be closely associated with the process of the epithelial-to-mesenchymal transition [24,25]. Multiple studies have also shown that L1CAM immunohistochemical expression has prognostic significance for endometrial cancer [21,22]. Moreover, it has been demonstrated that L1CAM expression is strongly associated with a high-grade histology or an abnormal p53 status [26,27].

In the immunohistochemical analysis of L1CAM in this study, we demonstrated that the rate of positivity for L1CAM expression was significantly lower in the EC^lo^p53^ab^ group than in the EC^hi^ group, while there was no significant difference in this variable between the EC^lo^p53^ab^ group and EC^lo^p53^wt^ group. Notably, unlike in the EC^hi^ group, there was no association between abnormal p53 expression and L1CAM expression in the EC^lo^p53^ab^ group.

Meanwhile, it is well recognized that p21 plays a key role in the regulation of the cell cycle and in DNA replication. In addition, Deiry et al. demonstrated that the induction of *CDKN1A* was associated with wild-type, but not mutant, *TP53* gene expression [28].

In the RNA sequencing analysis, *CDKN1A* was identified as being upregulated more frequently in the EC^lo^p53^ab^ group than in the EC^hi^ group and as being downregulated more frequently in the EC^lo^p53^ab^ group than in the EC^lo^p53^wt^ group. In the immunohistochemical analysis of p21 in this study, we demonstrated that the EC^lo^p53^wt^ group showed significantly more frequent p21-positive expression than the EC^lo^p53^ab^ group. However, the EC^lo^p53^ab^ group showed more frequent p21-positive expression than the EC^hi^ group, albeit not significantly.

We previously reported that G1 endometrioid carcinoma in the elderly is characterized by relatively aggressive behavior and occasional abnormal p53 expression, but the p53 status is not an independent predictor of a shorter PFS, and this is confined to G1 endometroid carcinoma [13].

There is a possibility that the difference in the expression of p21, which is associated with p53, affected the difference between the EC^lo^p53^ab^ group and EC^hi^ group, and also the difference between the EC^lo^p53^ab^ group and EC^lo^p53^wt^ group. Meanwhile, the Kyoto Encyclopedia of Genes and Genomes (KEGG) pathway analysis indicated that the DEGs were primarily enriched in pathways associated with cancer. In addition, *ESR1* encoding estrogen receptor alpha is included in these pathways, and the expression of ER may have influenced the differences between the EC^lo^p53^ab^ group and EC^hi^ group. Ogane et al. reported the ER (−) and p53 (+) pattern as an independent poor prognostic factor in endometrial cancer [29]. This is consistent with our results.

However, the RNA sequencing and immunohistochemical results from this study alone are not sufficient to investigate this difference, and it is necessary to increase the number of cases and include the null and cytoplasmic patterns of p53 expression.

In the survival analysis, the EC^lo^p53^ab^ group showed a longer progression-free and overall survival than the EC^hi^ group, which is consistent with our previous study [13]. In addition, Creasman et al. reported 5-year overall survival rates of 83.2% for endometrioid carcinoma and 52.6% for serous carcinoma [30]. In this study, the overall survival rate for the EC^lo^p53^ab^ group was 84.6%. Thus, we surmise that the condition in the EC^lo^p53^ab^ group was not as aggressive as that in the EC^hi^ group in this study.

As we mentioned previously, attempts have been made to apply the molecular classification of endometrial cancer clinically, as represented by the ESGO/ESTRO/ESP guidelines of 2021 and the FIGO 2023 staging [4,5]. If the molecular subtype is known, abnormal p53 endometrial cancer confined to the uterine corpus with any myometrial invasion, with or without cervical invasion, and regardless of the degree of lymphovascular invasion or the histological type, is classified as Stage 2Cm_p53abn_ in FIGO 2023 staging [5]. According to this new staging, conventional Stage 1 low-grade endometrial cancer with abnormal p53 expression would be upgraded to Stage 2Cm_p53abn_.

In addition, Yano et al. reported that the prognosis of low-grade endometrioid cancer with abnormal p53 expression is equivalent to that of high-grade endometrial cancer [11], and Puppo et al. reported that patients with low-grade endometrial cancer who exhibit abnormal p53 expression appear to have a higher risk of recurrence [12]. However, Vrede et al. reported that low-grade endometrial cancer has an excellent prognosis independent of the molecular subgroup, which does not support routine molecular profiling in patients with low-grade endometrial cancer [6]. In this way, conflicting findings on the clinical significance of abnormal p53 expression in low-grade endometrial cancer have been reported. However, if the treatment for the abnormal p53 expression group is modified, there is room for discussion as to whether this modification should be made for low-grade endometrial cancer. Our results suggest that p53 classification in low-grade endometrial cancer may not be necessary, but further investigation, such as larger multicenter studies, is necessary in this regard.

Similarly, our results suggest that the EC^lo^p53^ab^ group is a separate entity from the EC^hi^ group. However, we should mention the possibility that our cases in the EC^lo^p53^ab^ group were actually misclassified high-grade endometrial cancer. To rule out this possibility, via a careful review of HE staining, we confirmed that all cases in the EC^lo^p53^ab^ group in our study lacked high-grade nuclear atypia, exhibited smooth luminal borders, and retained diffuse ER expression. High-grade endometrial cancer, represented as serous carcinoma, is well known to be characterized by high-grade nuclear atypia and decreased ER expression. Thus, we concluded that our cases in the EC^lo^p53^ab^ group were not histologically or immunohistochemically misclassified high-grade endometrial cancer.

Moreover, in the molecular classification of endometrial cancer, MMR protein immunohistochemistry and a *POLE* mutation analysis should be performed prior to p53 immunohistochemistry [3,15,31]. Singh et al. reported that optimized p53 immunohistochemistry serves as an effective surrogate test for *TP53* mutations in endometrial cancer and shows an excellent reproducibility across laboratories [31]. In this study, all cases in the EC^lo^p53^ab^ group showed retained PMS2 and MSH6 immunohistochemical expression. In addition, all cases in the EC^lo^p53^ab^ group showed no *POLE* mutations in exons 9 and 13, as revealed by Sanger sequencing or RNA sequencing [13]. It has been reported that a small subset (3%–5%) of endometrial cancer patients show more than one alteration in the values for molecular classification, and endometrial cancer with a *POLE* mutation and abnormal p53 expression frequently showed subclonal mutant-like p53 expression [20,32]. Subclonal expression is defined as abrupt and complete regional abnormal p53 expression, in which the subclonal region is at least 10% of the tumor volume, but subclonal expression was not observed in the EC^lo^p53^ab^ group in this study. Therefore, based on the molecular classification, these patients were accurately assigned to the abnormal p53 group.

The ESGO/ESTRO/ESP guidelines of 2021 and the FIGO 2023 staging combine conventional staging with molecular classification based on comprehensive genomic and transcriptomic analyses of endometrial cancers and attempt to risk-stratify them. However, as reported by McCluggage et al., there is still inadequate evidence that low-grade endometrial cancer with abnormal p53 expression is equivalent to high-grade endometrial cancer [33]. Jamieson et al. demonstrated that abnormal p53 low-grade endometrial cancers carry a substantial risk of disease recurrence [34]. However, their study included low-grade endometrial cancer with decreased ER expression because a 10% cut-off for positivity was used for the assessment of ER expression. Estrogen dependency is one of the most important characteristics of low-grade endometrial endometrioid carcinoma [1], and further studies are needed to determine the difference between abnormal p53 low-grade endometrial cancer with retained diffuse ER expression, like the EC^lo^p53^ab^ group in this study, and abnormal p53 low-grade endometrial cancer with decreased ER expression.

## 5. Conclusions

The results of this study suggest that the EC^lo^p53^ab^ group is a separate entity from the EC^hi^ group, and that the EC^lo^p53^ab^ group is more similar to the EC^lo^p53^wt^ group than to the EC^hi^ group. There may be no need to routinely perform p53 immunohistochemistry in low-grade endometrial cancer, and if treatment for the abnormal p53 expression group is modified, it may not be necessary for low-grade endometrial cancer.

However, this study has a limitation associated with its small sample size. In this study, we limited the low-grade endometrial cancer group to cases with diffuse ER expression, so the number of cases in each subgroup is not sufficient to draw strong conclusions. To overcome this, larger multicenter studies that include more cases of low-grade endometrial cancer are needed. In addition, this study was a retrospective study, and a bias is that the abnormal p53 expression pattern in the EC^lo^p53^ab^ group only involved diffuse nuclear staining. For the adequate clinical application of the molecular classification in the future, more prospective evidence is needed to add other abnormal expression patterns of p53.

## Figures and Tables

**Figure 1 diagnostics-15-00671-f001:**
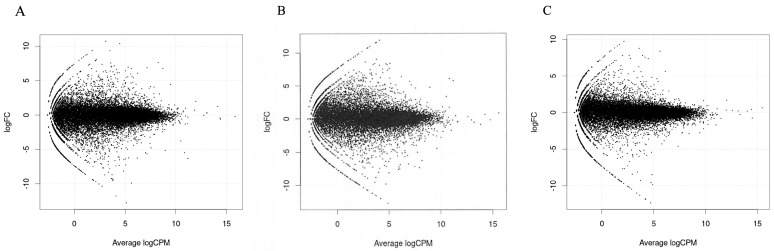
MA plots of RNA sequencing results comparing logFC (fold change) and logCPM (counts per million) were created. (**A**) EC^lo^p53^ab^ group and EC^hi^ group. (**B**) EC^lo^p53^wt^ group and EC^hi^ group. (**C**) EC^lo^p53^ab^ group and EC^lo^p53^wt^ group.

**Figure 2 diagnostics-15-00671-f002:**
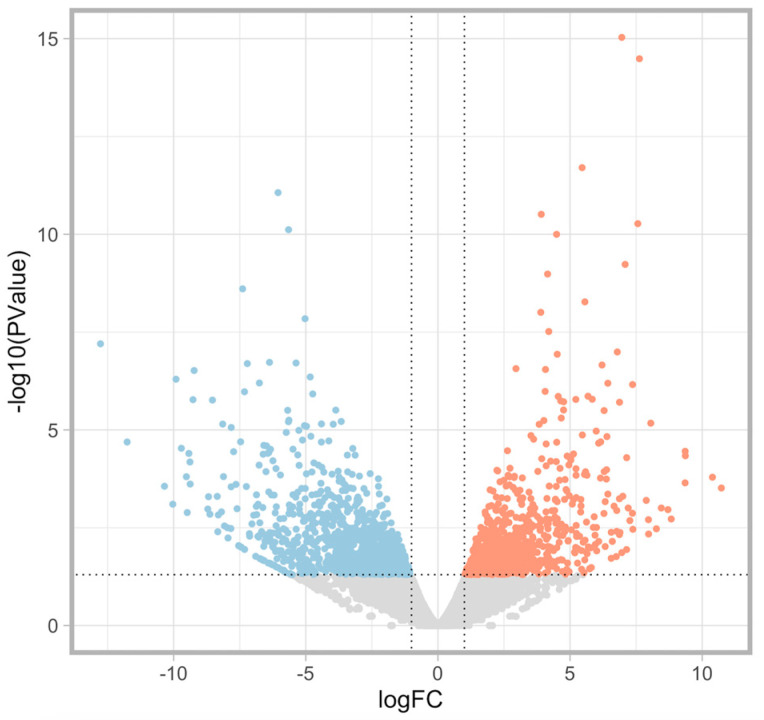
A volcano plot of differential gene expression patterns of the EC^lo^p53^ab^ group versus the EC^hi^ group to identify specific genes that fit the threshold criteria (*p*-value < 0.05 and logFC ≥ 1 or ≤−1). The differentially expressed genes (DEGs) are shown in blue (downregulated) or red (upregulated).

**Figure 3 diagnostics-15-00671-f003:**
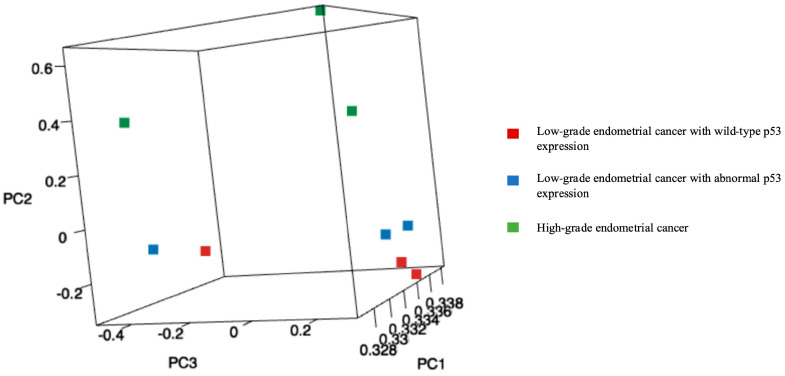
Principal component analysis (PCA): A plot of the first three principal components (PC1, PC2, and PC3). In the PCA, the EC^lo^p53^ab^ group was more similar to the EC^lo^p53^wt^ group than to the EC^hi^ group (red squares: low-grade endometrial cancer with wild-type p53 expression, blue squares: low-grade endometrial cancer with abnormal p53 expression, and green squares: high-grade endometrial cancer).

**Figure 4 diagnostics-15-00671-f004:**
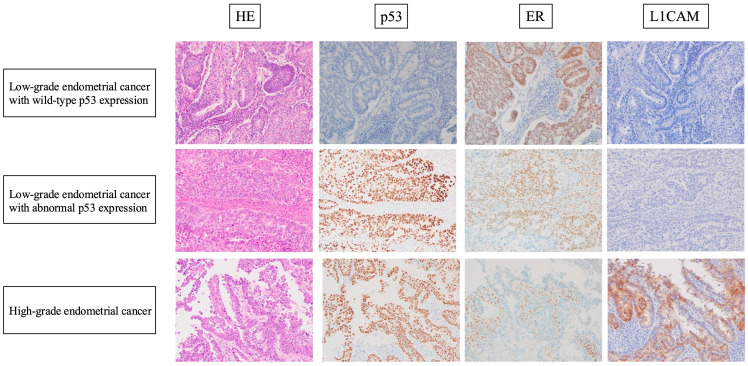
Representative HE-stained and immunohistochemical images of cases of each of the three groups. Low-grade nuclear atypia, smooth luminal borders, and irregular glands or a cribriform pattern are shown in both the EC^lo^p53^ab^ group and the EC^lo^p53^wt^ group. Retained ER expression and negativity for L1CAM expression were also observed in the tumor cells in the two EC^lo^ groups. High-grade nuclear atypia and a papillary pattern are shown in the EC^hi^ group. ER loss, abnormal p53 expression, and positivity for L1CAM expression were observed in the tumor cells of the EC^hi^ group.

**Figure 5 diagnostics-15-00671-f005:**
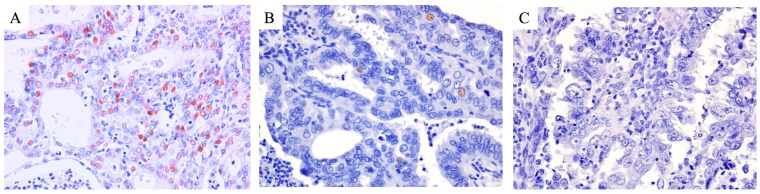
Representative p21 immunohistochemical images of cases of each of the three groups. The nuclear expression of p21 was diffusely observed in the tumor cells of the EC^lo^p53^wt^ group (**A**) and focally observed in the tumor cells of the EC^lo^p53^ab^ group (**B**). Meanwhile, p21 loss (no nuclear expression) was observed in the tumor cells of the EC^hi^ group (**C**).

**Figure 6 diagnostics-15-00671-f006:**
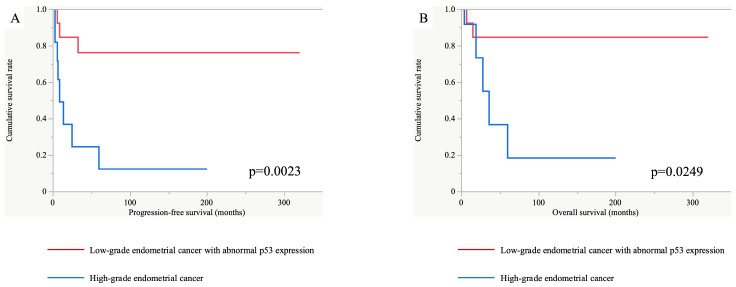
Progression-free and overall survival analyses between the EC^lo^p53^ab^ group (red line) and EC^hi^ group (blue line). The progression-free (**A**) and overall (**B**) survival periods were significantly longer in the EC^lo^p53^ab^ group than in the EC^hi^ group (*p* = 0.0023 and 0.0249, respectively).

**Table 1 diagnostics-15-00671-t001:** Clinicopathological characteristics of the 44 patients with endometrial cancer.

	Low-Grade Endometrial Cancer with Wild-Type p53 Expression (n = 16)	Low-Grade Endometrial Cancer with Abnormal p53 Expression (n = 13)	High-Grade Endometrial Cancer (n = 15)
Age, n (%)			
<60 years	5 (31.3%)	4 (30.8%)	2 (13.3%)
≥60 years	11 (68.7%)	9 (69.2%)	13 (86.7%)
Myometrial invasion, n (%)			
<1/2	4 (25.0%)	9 (69.2%)	5 (33.3%)
≥1/2	12 (75.0%)	4 (30.8%)	10 (66.7%)
FIGO 2023 stage, n (%)			
Early (I+II)	12 (75.0%)	10 (76.9%)	5 (33.3%)
Advanced (III+IV)	4 (25.0%)	3 (23.1%)	10 (66.7%)
Distant metastasis, n (%)			
Present	1 (6.3%)	1 (7.7%)	5 (33.3%)
Absent	15 (93.7%)	12 (92.3%)	10 (66.7%)
Lymph node metastasis, n (%)			
Present (n = 9)			
ITCs	1 (6.7%)	0 (0%)	0 (0%)
Micro-metastasis	2 (13.3%)	0 (0%)	2 (18.2%)
Macro-metastasis	1 (6.7%)	1 (8.3%)	2 (18.2%)
Absent (n = 29)	11 (73.3%)	11 (91.7%)	7 (63.6%)
Uterine cervical invasion, n (%)			
Present	2 (12.5%)	2 (15.4%)	2 (13.3%)
Absent	14 (87.5%)	11 (84.6%)	13 (86.7%)
Peritoneal cytology, n (%)			
Positive	1 (6.3%)	1 (7.7%)	8 (53.5%)
Negative	15 (93.7%)	12 (92.3%)	7 (46.7%)
Lymphovascular invasion, n (%)			
None	6 (37.5%)	10 (76.9%)	7 (46.7%)
Focal	4 (25.0%)	2 (15.4%)	3 (20.0%)
Substantial	6 (37.5%)	1 (7.7%)	5 (33.3%)
Hysterectomy, n (%)			
Yes	16 (100%)	13 (100%)	15 (100%)
No	0 (0%)	0 (0%)	0 (0%)
Lymphadenectomy, n (%)			
Yes	15 (93.8%)	13 (100%)	13 (86.7%)
No	1 (6.2%)	0 (0%)	2 (13.3%)
Recurrence, n (%)			
Yes (n = 10)	0 (0%)	3 (23.1%)	7 (58.3%)
No (n = 31)	16 (100%)	10 (76.9%)	5 (41.7%)

FIGO: International Federation of Gynecology and Obstetrics.

**Table 2 diagnostics-15-00671-t002:** L1CAM status and L1CAM immunohistochemical scoring for each of the three groups.

	Low-Grade Endometrial Cancer with Wild-Type p53 Expression (n = 16)	Low-Grade Endometrial Cancer with Abnormal p53 Expression (n = 13)	High-Grade Endometrial Cancer (n = 15)
L1CAM immunohistochemical scoring, n (%)			
0	10 (62.5%)	7 (53.8%)	0 (0%)
1	6 (37.5%)	4 (30.8%)	1 (6.7%)
2	0 (0%)	1 (7.7%)	5 (33.3%)
3	0 (0%)	1 (7.7%)	9 (60.0%)
L1CAM expression, n (%)			
positive	0 (0%)	2 (15.4%) *	14 (93.3%) *
negative	16 (100%)	11 (84.6%) *	1 (6.7%) *

* *p* < 0.05, low-grade endometrial cancer with abnormal p53 expression versus high-grade endometrial cancer.

**Table 3 diagnostics-15-00671-t003:** p21 immunohistochemical expression for each of the three groups.

	Low-Grade Endometrial Cancer with Wild-Type p53 Expression (n = 16)	Low-Grade Endometrial Cancer with Abnormal p53 Expression (n = 13)	High-Grade Endometrial Cancer (n = 15)
p21 immunohistochemical expression, n (%)			
positive	13 (81.3%) *	6 (46.2%) *	4 (26.7%)
negative	3 (18.7%) *	7 (53.8%) *	11 (73.3%)

* *p* < 0.05, low-grade endometrial cancer with wild-type p53 expression versus low-grade endometrial cancer with abnormal p53 expression.

## Data Availability

The datasets generated and/or analyzed during the current study are available in the Gene Expression Omnibus (GEO) repository, GSE275015.

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
