# Peer review of "Low-Grade Endometrial Cancer with Abnormal p53 Expression as a Separate Clinical Entity: Insights from RNA Sequencing and Immunohistochemistry"

_diagnostics, 2025, doi:10.3390/diagnostics15060671_

Round 1
Reviewer 1 Report
Comments and Suggestions for Authors
Dear Authors,
Congratulations for your study.
The title is informative, but it could be more specific by indicating the methodology (RNA sequencing and immunohistochemistry).
The abstract effectively summarizes the findings but would benefit from a clearer statement on clinical implications and the potential impact of these results.
Consider rewording the conclusion in the abstract to emphasize the distinction between p53-abnormal low-grade endometrial cancer and high-grade endometrial cancer, as this distinction is the main finding.
The introduction provides a good background but could be improved by including more recent references on the role of p53 in endometrial cancer.
The rationale for the study is not entirely clear. Why is it important to differentiate these subtypes in clinical practice? Clarifying this would strengthen the introduction.
The study methodology is well described, but additional information on sample size justification would be beneficial. How was the sample size determined? Were power calculations performed?
The description of RNA sequencing should include information on quality control steps beyond RNA integrity.
It would be helpful to clarify how the immunohistochemical scoring was validated. Were multiple pathologists involved in scoring to ensure interobserver reliability?
The presentation of RNA sequencing results is clear, but including volcano plots or additional visualizations of differentially expressed genes would enhance clarity.
The immunohistochemistry results are well presented, but more details on how p53 staining patterns were classified would improve reproducibility.
The survival analysis is compelling, but more detailed hazard ratios and confidence intervals should be included in the statistical reporting.
The discussion does a good job of contextualizing the results, but there are areas where additional discussion would be valuable.
How do the findings compare to other recent studies on p53 in endometrial cancer?
What are the potential therapeutic implications of identifying this distinct subgroup?
You mention that the findings suggest a distinct clinical entity, but there is limited discussion on how this could change clinical management. Could this impact treatment decisions or follow-up recommendations?
More acknowledgment of study limitations is needed. For example: The small sample size may limit the generalizability of the findings, the retrospective nature of the study could introduce bias, how does interlaboratory variability in p53 immunohistochemistry impact the reproducibility of these findings?
The conclusion should be more specific in outlining the next steps for research. Should larger multicenter studies be conducted? Should this classification be tested prospectively?
Clarify whether these findings support a change in current clinical guidelines.
Kind regards.
Author Response
Responses to the reviewers' comments
Diagnostics
Manuscript. No.: diagnostics-3510834
Title: Low-grade endometrial cancer with abnormal p53 expression as a separate clinical entity
Corresponding Author: Kazuhisa Hachisuga
Dear Professor Andreas Kjaer,
We are very grateful for your constructive comments and for the time taken to review our manuscript.
Based on your comments, we have revised our manuscript, as follows.
Reviewer #1:
1-The title is informative, but it could be more specific by indicating the methodology (RNA sequencing and immunohistochemistry).
Reply:
We agree with the reviewer, and the title has been changed as follows:
Low-grade endometrial cancer with abnormal p53 expression as a separate clinical entity: Insights from RNA sequencing and immunohistochemistry.
2-The abstract effectively summarizes the findings but would benefit from a clearer statement on clinical implications and the potential impact of these results.
Reply:
We agree with the reviewer , and we have added the following sentences in the revised manuscript.
Page 2, line 43-45
While abnormal p53 group is considered the most prognostically unfavorable in molecular classification, these findings suggest that routine molecular profiling is not necessary for patients with low-grade endometrial cancer. However, there is insufficient evidence to modify adjuvant treatment in low-grade endometrial cancer patients.
3- Consider rewording the conclusion in the abstract to emphasize the distinction between p53-abnormal low-grade endometrial cancer and high-grade endometrial cancer, as this distinction is the main finding.
Reply:
We agree with the reviewer , and we have added the following sentences in the revised manuscript.
Page 2, line 43-45
While abnormal p53 group is considered the most prognostically unfavorable in molecular classification, these findings suggest that routine molecular profiling is not necessary for patients with low-grade endometrial cancer.
4- The introduction provides a good background but could be improved by including more recent references on the role of p53 in endometrial cancer.
Reply:
We agree with the reviewer , and we have added the following sentences and reference in the revised manuscript.
Page 4, line75-77
Meanwhile, Puppo et al. demonstrated conflict findings that low-grade endometrial cancer patients with abnormal p53 expression seem to be at greater risk of recurrence [12].
- Puppo, A.; Fraternali Orcioni, G.; Clignon, V.; Musizzano, Y.; Zavattero, C.A.; Vocino Trucco, G.; Benazzo, G.M.; Vizzielli, G.; Restaino, S.; Mariuzzi, L.; et al. Where Morphological and Molecular Classifications Meet: The Role of p53 Immunohistochemistry in the Prognosis of Low-Risk Endometrial Carcinoma (GLAMOUR Study). Cancers (Basel) 2024, 16.
5-The rationale for the study is not entirely clear. Why is it important to differentiate these subtypes in clinical practice? Clarifying this would strengthen the introduction.
Reply:
We agree with the reviewer, and we have added the following sentences in the revised manuscript.
Page4, line 77-80
When molecular classification is applied to low-grade endometrial cancer, there is a possibility that adjuvant therapy may be modified for the abnormal p53 group. However, there is still insufficient evidence to support this. The question of whether to evaluate p53 expression in all cases of low-grade endometrial cancer also warrants further investigation.
6- The study methodology is well described, but additional information on sample size justification would be beneficial. How was the sample size determined? Were power calculations performed?
Reply:
We agree with the reviewer, and we have added the following sentences as a limitation for the sample size in the revised manuscript.
Page 15, line 352-355
However, this study has a limitation associated with its small sample size. In this study, we limited the low-grade endometrial cancer group to cases with diffuse ER expression, so the number of cases in each subgroup is not sufficient to draw strong conclusions. To overcome this, larger multicenter studies including more cases of low-grade endometrial cancer are needed.
7-The description of RNA sequencing should include information on quality control steps beyond RNA integrity.
Reply:
We agree with the reviewer, and we have added the following sentences in the revised manuscript.
Page 5-6, line 119-120
All RNA used for sequencing was confirmed that there RIN value were 7.0 or higher.
8- It would be helpful to clarify how the immunohistochemical scoring was validated. Were multiple pathologists involved in scoring to ensure interobserver reliability?
Reply:
We agree with the reviewer, and we have added the following sentences about the immunohistochemical scoring in the revised manuscript.
Page 7, line 163-165
The evaluations of immunohistochemical p53 expression were performed by one pathologist and one physician, who specialize in gynecological oncology and were blinded to the patients’ clinicopathological details [11].
9-The presentation of RNA sequencing results is clear, but including volcano plots or additional visualizations of differentially expressed genes would enhance clarity.
Reply:
We agree with the reviewer, and we have added the volcano plot of the EClop53ab group versus the EChi group and the following sentences in the revised manuscript.
Page 9, line 194-196
The volcano plot of differential gene expression patterns of the EClop53ab group versus the EChi group was also created (Figure 2).
Page 20, line 509-512
Figure 2. The volcano plot of differential gene expression patterns of the EClop53ab group versus the EChi group to identify specific genes that fit the threshold criteria (p-value < 0.05 and logFC ≥ 1 or ≤ -1). The differentially expressed genes (DEGs) were shown in blue (downregulated) or red (upregulated).
10-The immunohistochemistry results are well presented, but more details on how p53 staining patterns were classified would improve reproducibility.
Reply:
We agree with the reviewer, and we have added the following sentences in the revised manuscript.
Page 7, page 158-163
Abnormal p53 expression was defined as strong nuclear staining in at least 80% of tumor cell nuclei, or the absence of staining in tumor cell nuclei when the "wild-type" pattern was present, which ranged from a few positive cells to almost all cells being stained, though with varying intensity [11,12]. The cytoplasmic pattern of p53 expression was characterized by predominant cytoplasmic staining without strong nuclear staining in more than 80% of tumor cell nuclei [12,20].
11- The survival analysis is compelling, but more detailed hazard ratios and confidence intervals should be included in the statistical reporting.
Reply:
We agree with the reviewer, and we have added the Cox proportional hazards model and following sentences in the revised manuscript.
Page 8, line 175-176
Cox proportional hazards model was performed for univariate survival analyses.
Page 10, line 230-233
In addition, univariate analyses demonstrated that the hazard ratios for PFS and OS in the EChi group versus the EClop53ab group were 6.30 (95% confidence interval, 1.62-24.47) and 5.66 (95% confidence interval, 1.05-30.47), respectively.
12- The discussion does a good job of contextualizing the results, but there are areas where additional discussion would be valuable.
Reply:
We agree with the reviewer, and we have added the following sentences and reference in the revised manuscript.
Page 12, line 275-280
Meanwhile, the Kyoto Encyclopedia of Genes and Genomes (KEGG) pathway analysis indicated that the DEGs were primarily enriched in pathways associated with cancer. In addition, ESR1 encoding estrogen receptor-alpha is also included in these pathways, and the expression of ER, may be influencing the differences between the EClop53ab group and EChi group. Ogane et al. reported that the ER (-) and p53 (+) pattern as an independent poor prognostic factor in endometrial cancer [29]. It is consistent with our results.
- Ogane, N.; Hori, S.I.; Yano, M.; Katoh, T.; Kamoshida, S.; Kato, H.; Kameda, Y.; Yasuda, M. Preponderance of endometrial carcinoma in elderly patients. Mol Clin Oncol 2018, 9, 269-273.
13- How do the findings compare to other recent studies on p53 in endometrial cancer?
Reply:
We agree with the reviewer, and we have added the following sentences in the revised manuscript.
Page 13, line 297-308
In addition, Yano et al. reported that the prognosis of low-grade endometrioid cancer with abnormal p53 expression is equivalent to that of high-grade endometrial cancer [11] and Puppo et al. reported that the patients with low-grade endometrial cancer who exhibit abnormal p53 expression appear to have a higher risk of recurrence [12]. However, Vrede et al. reported that low-grade endometrial cancer had an excellent prognosis independent of molecular subgroup, which does not support routine molecular profiling in patients with low-grade endometrial cancer [6]. In this way, conflict findings on the clinical significance of p53 abnormal expression in low-grade endometrial cancer were reported. However, if treatment for the abnormal p53 expression group is modified, there is room for discussion as to whether this modifications should be made for low-grade endometrial cancer. Our results suggest that p53 classification in low-grade endometrial cancer may not be necessary, but further investigation, such as larger multicenter studies, is necessary in this regard.
14- What are the potential therapeutic implications of identifying this distinct subgroup?
Reply:
We agree with the reviewer, and we have added the following sentences in the revised manuscript.
Page 13, line 304-308
However, if treatment for the abnormal p53 expression group is modified, there is room for discussion as to whether this modifications should be made for low-grade endometrial cancer. Our results suggest that p53 classification in low-grade endometrial cancer may not be necessary, but further investigation, such as larger multicenter studies, is necessary in this regard.
15- You mention that the findings suggest a distinct clinical entity, but there is limited discussion on how this could change clinical management. Could this impact treatment decisions or follow-up recommendations?
Reply:
We agree with the reviewer, and we have added the following sentences in the revised manuscript.
Page 13, line 304-308
However, if treatment for the abnormal p53 expression group is modified, there is room for discussion as to whether this modifications should be made for low-grade endometrial cancer. Our results suggest that p53 classification in low-grade endometrial cancer may not be necessary, but further investigation, such as larger multicenter studies, is necessary in this regard.
16- More acknowledgment of study limitations is needed. For example: The small sample size may limit the generalizability of the findings, the retrospective nature of the study could introduce bias, how does interlaboratory variability in p53 immunohistochemistry impact the reproducibility of these findings?
Reply:
We agree with the reviewer, and we have added the following sentences in the revised manuscript. Regarding the interlaboratory variability in p53 immunohistochemistry, details on classification of p53 staining patterns were indicated to improve reproducibility.
Page 15, line 355-357
In addition, this study is a retrospective study, and a bias is that the abnormal p53 expression pattern in the EClop53ab group only involved diffuse nuclear staining.
Page 15, line 352-355
However, this study has a limitation associated with its small sample size. In this study, we limited the low-grade endometrial cancer group to cases with diffuse ER expression, so the number of cases in each subgroup is not sufficient to draw strong conclusions. To overcome this, larger multicenter studies including more cases of low-grade endometrial cancer are needed.
Page 7, line 158-163
Abnormal p53 expression was defined as strong nuclear staining in at least 80% of tumor cell nuclei, or the absence of staining in tumor cell nuclei when the "wild-type" pattern was present, which ranged from a few positive cells to almost all cells being stained, though with varying intensity [11,12]. The cytoplasmic pattern of p53 expression was characterized by predominant cytoplasmic staining without strong nuclear staining in more than 80% of tumor cell nuclei [12,20].
17- The conclusion should be more specific in outlining the next steps for research. Should larger multicenter studies be conducted? Should this classification be tested prospectively?
Reply:
We agree with the reviewer, and we have added the following sentences.
Page 15, line 348-351
There may be no need to routinely perform p53 immunohistochemistry in low-grade endometrial cancer, and if treatment for the abnormal p53 expression group is modified, it may not be necessary for low-grade endometrial cancer.
Page 15, line 354-355
To overcome this, larger multicenter studies including more cases of low-grade endometrial cancer are needed.
18- Clarify whether these findings support a change in current clinical guidelines.
Reply:
We agree with the reviewer, and we have added the following sentences in the revised manuscript. To reconsider guidelines, further investigation, including large multicenter studies, is necessary.
Page 15, line 348-351
There may be no need to routinely perform p53 immunohistochemistry in low-grade endometrial cancer, and if treatment for the abnormal p53 expression group is modified, it may not be necessary for low-grade endometrial cancer.
Reviewer 2 Report
Comments and Suggestions for Authors
Here are my comments to improve the scientific content of your manuscript:
The introduction provides a strong background but does not explicitly state why investigating low-grade endometrial cancer with abnormal p53 expression is crucial. Consider adding a statement on the clinical relevance and potential impact on treatment decisions.
You mention the TCGA molecular classification, but its direct applicability to low-grade endometrial cancer remains vague. Provide a clearer rationale for why molecular classification might or might not be useful in this setting.
The study has a small sample size (n=9 for RNA-seq and n=44 for IHC). Consider acknowledging this more explicitly and discussing potential biases or how future studies could overcome this limitation.
Explain in greater detail how abnormal p53 cases were identified. Was p53 sequencing performed, or was classification solely based on immunohistochemistry? This is crucial for reproducibility.
Provide more details on statistical power calculations. Were the sample sizes sufficient to detect meaningful differences?
The PCA suggests that the EClop53ab group is more similar to the EClop53wt group than the EChi group, yet differential expression analysis shows substantial differences (1811 DEGs vs. high-grade, 1088 vs. wild-type). Clarify how these findings reconcile—does this imply partial overlap, or do the key differences indicate a distinct subgroup?
Discuss the significance of key DEGs beyond L1CAM and CDKN1A. What pathways were enriched in these comparisons? Would additional functional validation (e.g., pathway analysis, gene ontology) strengthen the conclusions?
You note that L1CAM is significantly different between EClop53ab and EChi but not between EClop53ab and EClop53wt. This weakens its case as a distinct marker. Consider discussing whether alternative biomarkers might better differentiate EClop53ab tumors.
Some prior studies (e.g., Yano et al., Vrede et al.) are mentioned but not critically discussed in terms of agreement or contradiction with your findings. Do your results support the idea that p53 abnormalities in low-grade EC are clinically significant, or do they align with the argument that molecular classification is unnecessary in this setting?
If EClop53ab is a separate entity, how should it be treated? Would molecular classification affect treatment decisions (e.g., adjuvant therapy, monitoring strategies)? This is a crucial missing piece in the discussion.
Could the differences in p21 expression be due to variations in TP53 mutation types (e.g., gain-of-function vs. loss-of-function mutations)? Discuss how sequencing data could refine classification.
The conclusion states that EClop53ab is a separate entity but does not clarify how this should influence clinical practice. Should guidelines be reconsidered?
Expand on what future work is necessary to confirm these findings (e.g., larger cohorts, functional studies, treatment response data).
Author Response
Responses to the reviewers' comments
Diagnostics
Manuscript. No.: diagnostics-3510834
Title: Low-grade endometrial cancer with abnormal p53 expression as a separate clinical entity
Corresponding Author: Kazuhisa Hachisuga
Dear Professor Andreas Kjaer,
We are very grateful for your constructive comments and for the time taken to review our manuscript.
Based on your comments, we have revised our manuscript, as follows.
Reviewer2
1- The introduction provides a strong background but does not explicitly state why investigating low-grade endometrial cancer with abnormal p53 expression is crucial. Consider adding a statement on the clinical relevance and potential impact on treatment decisions.
Reply:
We agree with the reviewer, and we have added the following sentences in the revised manuscript.
Page 4, line 75-80
Meanwhile, Puppo et al. demonstrated conflict findings that low-grade endometrial cancer patients with abnormal p53 expression seem to be at greater risk of recurrence [12]. When molecular classification is applied to low-grade endometrial cancer, there is a possibility that adjuvant therapy may be modified for the abnormal p53 group. However, there is still insufficient evidence to support this. The question of whether to evaluate p53 expression in all cases of low-grade endometrial cancer also warrants further investigation.
2- You mention the TCGA molecular classification, but its direct applicability to low-grade endometrial cancer remains vague. Provide a clearer rationale for why molecular classification might or might not be useful in this setting.
Reply:
We agree with the reviewer, and we have added the following sentences in the revised manuscript.
Page 3, line 70-71
Vrede et al. reported that low-grade endometrial cancer has an excellent prognosis, regardless of its molecular subgroup [6].
3- The study has a small sample size (n=9 for RNA-seq and n=44 for IHC). Consider acknowledging this more explicitly and discussing potential biases or how future studies could overcome this limitation.
Reply:
We agree with the reviewer, and we have added the following sentences in the revised manuscript.
Page 15, line 352-355
However, this study has a limitation associated with its small sample size. In this study, we limited the low-grade endometrial cancer group to cases with diffuse ER expression, so the number of cases in each subgroup is not sufficient to draw strong conclusions. To overcome this, larger multicenter studies including more cases of low-grade endometrial cancer are needed.
4- Explain in greater detail how abnormal p53 cases were identified. Was p53 sequencing performed, or was classification solely based on immunohistochemistry? This is crucial for reproducibility.
Reply:
The classification of p53 expression was based on immunohistochemistry in this study. Regarding reproducibility, we have added the following sentences in the revised manuscript.
Page 14, line 320-322
Singh et al. reported that optimized p53 immunohistochemistry serves as an effective surrogate test for TP53 mutations in endometrial cancer and shows excellent reproducibility across laboratories [31].
Page 7, line 158-163
Abnormal p53 expression was defined as strong nuclear staining in at least 80% of tumor cell nuclei, or the absence of staining in tumor cell nuclei when the "wild-type" pattern was present, which ranged from a few positive cells to almost all cells being stained, though with varying intensity [11,12]. The cytoplasmic pattern of p53 expression was characterized by predominant cytoplasmic staining without strong nuclear staining in more than 80% of tumor cell nuclei [12,20].
5- Provide more details on statistical power calculations. Were the sample sizes sufficient to detect meaningful differences?
Reply:
As pointed out by the reviewer, we have added the following sentences in the revised manuscript.
Page 15, line 352-355
However, this study has a limitation associated with its small sample size. In this study, we limited the low-grade endometrial cancer group to cases with diffuse ER expression, so the number of cases in each subgroup is not sufficient to draw strong conclusions. To overcome this, larger multicenter studies including more cases of low-grade endometrial cancer are needed.
6-The PCA suggests that the EClop53ab group is more similar to the EClop53wt group than the EChi group, yet differential expression analysis shows substantial differences (1811 DEGs vs. high-grade, 1088 vs. wild-type). Clarify how these findings reconcile—does this imply partial overlap, or do the key differences indicate a distinct subgroup?
Reply:
As pointed out by the reviewer, we demonstrated that the EClop53ab group was more similar to the EClop53wt group than to the EChigroup in PCA. In addition, we demonstrated that DEGs were identified in the RNA-sequencing results (1811 genes between the EClop53ab group and EChi group, 1088 genes between the EClop53ab group and EClop53wt group).
The fact that more DEGs were identified between the EClop53ab group and EChi group compared to the DEGs identified between the EClop53ab group and EClop53wt group is consistent with the PCA results.
Additionally, this study focuses on CDKN1A, a gene associated with p53, which is recognized as a DEG in each comparison as shown below:
Page 9, line 203-207
In RNA-sequencing analysis comparing the EClop53ab group to the EChi group, L1CAM was identified as a downregulated gene in the former group, while CDKN1A encoding p21 was upregulated in it. In addition, in RNA-sequencing analysis comparing the EClop53ab group to the EClop53wt group, CDKN1A was identified as a downregulated gene in the former group. Both L1CAM and p21 expression are associated with p53 expression.
7- Discuss the significance of key DEGs beyond L1CAM and CDKN1A. What pathways were enriched in these comparisons? Would additional functional validation (e.g., pathway analysis, gene ontology) strengthen the conclusions?
Page 12, line 275-280
Meanwhile, the Kyoto Encyclopedia of Genes and Genomes (KEGG) pathway analysis indicated that the DEGs were primarily enriched in pathways associated with cancer. In addition, ESR1 encoding estrogen receptor-alpha is also included in these pathways, and the expression of ER, may be influencing the differences between the EClop53ab group and EChi group. Ogane et al. reported that the ER (-) and p53 (+) pattern as an independent poor prognostic factor in endometrial cancer [29]. It is consistent with our results.
8-You note that L1CAM is significantly different between EClop53ab and EChi but not between EClop53ab and EClop53wt. This weakens its case as a distinct marker. Consider discussing whether alternative biomarkers might better differentiate EClop53ab tumors.
Reply:
An important point in our study is that the EClop53ab group was more similar to the EClop53wt group than to the EChi group. Therefore, we believe L1CAM could be a valuable marker. However, as you pointed out, identifying markers that can distinguish each group may be useful, and this should also be investigated in future research.
9-Some prior studies (e.g., Yano et al., Vrede et al.) are mentioned but not critically discussed in terms of agreement or contradiction with your findings. Do your results support the idea that p53 abnormalities in low-grade EC are clinically significant, or do they align with the argument that molecular classification is unnecessary in this setting?
Reply:
We agree with the reviewer, and we have added the following sentences in the revised manuscript.
Page 13, line 303-308
In this way, conflict findings on the clinical significance of abnormal p53 expression in low-grade endometrial cancer were reported. However, if treatment for the abnormal p53 expression group is modified, there is room for discussion as to whether this modifications should be made for low-grade endometrial cancer. Our results suggest that p53 classification in low-grade endometrial cancer may not be necessary, but further investigation, such as larger multicenter studies, is necessary in this regard.
10-If EClop53ab is a separate entity, how should it be treated? Would molecular classification affect treatment decisions (e.g., adjuvant therapy, monitoring strategies)? This is a crucial missing piece in the discussion.
Reply:
We agree with the reviewer, and we have added the following sentences in the revised manuscript.
Page 13, line 304-308
However, if treatment for the abnormal p53 expression group is modified, there is room for discussion as to whether this modifications should be made for low-grade endometrial cancer. Our results suggest that p53 classification in low-grade endometrial cancer may not be necessary, but further investigation, such as larger multicenter studies, is necessary in this regard.
11-Could the differences in p21 expression be due to variations in TP53 mutation types (e.g., gain-of-function vs. loss-of-function mutations)? Discuss how sequencing data could refine classification.
Reply:
As pointed out by the reviewer, It is an interesting point what influences the differences in p21 expression. And so we have added the following sentences in the revised manuscript.
Page 12, line 275-279
Meanwhile, the Kyoto Encyclopedia of Genes and Genomes (KEGG) pathway analysis indicated that the DEGs were primarily enriched in pathways associated with cancer. In addition, ESR1 encoding estrogen receptor-alpha is also included in these pathways, and the expression of ER, may be influencing the differences between the EClop53ab group and EChi group.
Page 12, line 281-283
However, the RNA sequencing and immunohistochemical results from this study alone are not sufficient to investigate this difference, and it is necessary to increase the number of cases and include the null and cytoplasmic patterns of p53 expression.
12-The conclusion states that EClop53ab is a separate entity but does not clarify how this should influence clinical practice. Should guidelines be reconsidered?
Reply:
We agree with the reviewer, and we have added the following sentences in the revised manuscript. To reconsider guidelines, further investigation, including large multicenter studies, is necessary.
Page 15, line 348-351
There may be no need to routinely perform p53 immunohistochemistry in low-grade endometrial cancer, and if treatment for the abnormal p53 expression group is modified, it may not be necessary for low-grade endometrial cancer.
13-Expand on what future work is necessary to confirm these findings (e.g., larger cohorts, functional studies, treatment response data).
Reply:
As pointed out by the reviewer, we have added the following sentences in the revised manuscript.
Page 15, line 352-359
However, this study has a limitation associated with its small sample size. In this study, we limited the low-grade endometrial cancer group to cases with diffuse ER expression, so the number of cases in each subgroup is not sufficient to draw strong conclusions. To overcome this, larger multicenter studies including more cases of low-grade endometrial cancer are needed. In addition, this study is a retrospective study, and a bias is that the abnormal p53 expression pattern in the EClop53ab group only involved diffuse nuclear staining. For adequate clinical application of the molecular classification in the future, more prospective evidence is needed to add other abnormal expression patterns of p53.
Round 2
Reviewer 2 Report
Comments and Suggestions for Authors
The authors have addressed the concerns.
Comments on the Quality of English LanguageThe English language used is easy to understand.